# Metabolomic Profiling and Cytotoxic Tetrahydrofurofuran Lignans Investigations from *Premna odorata* Blanco

**DOI:** 10.3390/metabo9100223

**Published:** 2019-10-13

**Authors:** Abeer H. Elmaidomy, Rabab Mohammed, Hossam M. Hassan, Asmaa I. Owis, Mostafa E. Rateb, Mohammad A. Khanfar, Markus Krischke, Martin J. Mueller, Usama Ramadan Abdelmohsen

**Affiliations:** 1Department of Pharmacognosy, Faculty of Pharmacy, Beni-Suef University, Beni-Suef 62514, Egypt; abeerabdelhakium@yahoo.com (A.H.E.); rmwork06@yahoo.com (R.M.); abuh20050@yahoo.com (H.M.H.); mostafa19772002@yahoo.com (M.E.R.); 2School of Computing, Engineering & Physical Sciences, University of the West of Scotland, Paisley PA1 2BE, UK; 3Faculty of Pharmacy, The University of Jordan, P.O Box 13140, Amman 11942, Jordan; m_khanfar@ju.edu.jo; 4College of Pharmacy, Alfaisal University, Al Takhassusi Rd, Riyadh 11533, Saudi Arabia; 5Julius-von-Sachs-Institute of Biosciences, Biocenter, Pharmaceutical Biology, University of Würzburg, 97070 Würzburg, Germany; krischke@biozentrum.uni-wuerzburg.de; 6Department of Pharmacognosy, Faculty of Pharmacy, Deraya University, Universities Zone, New Minia City 61111, Egypt; 7Department of Pharmacognosy, Faculty of Pharmacy, Minia University, Minia 61519, Egypt

**Keywords:** *Premna*, lignan, metabolomic, cytotoxic, pharmacophore map

## Abstract

Metabolomic profiling of different *Premna odorata* Blanco (Lamiaceae) organs, bark, wood, young stems, flowers, and fruits dereplicated 20, 20, 10, 20, and 20 compounds, respectively, using LC–HRESIMS. The identified metabolites (**1**–**34**) belonged to different chemical classes, including iridoids, flavones, phenyl ethanoids, and lignans. A phytochemical investigation of *P. odorata* bark afforded one new tetrahydrofurofuran lignan, 4*β*-hydroxyasarinin **35**, along with fourteen known compounds. The structure of the new compound was confirmed using extensive 1D and 2D NMR, and HRESIMS analyses. A cytotoxic investigation of compounds **35**–**38** against the HL-60, HT-29, and MCF-7 cancer cell lines, using the MTT assay showed that compound **35** had cytotoxic effects against HL-60 and MCF-7 with IC_50_ values of 2.7 and 4.2 µg/mL, respectively. A pharmacophore map of compounds **35** showed two hydrogen bond acceptor (HBA) aligning the phenoxy oxygen atoms of benzodioxole moieties, two aromatic ring features vectored on the two phenyl rings, one hydrogen bond donor (HBD) feature aligning the central hydroxyl group and thirteen exclusion spheres which limit the boundaries of sterically inaccessible regions of the target’s active site.

## 1. Introduction

Cancer is one of the most threatening diseases worldwide that affect human health and quality of life [1,2,3]. Chemotherapy is frequently used in cancer treatment, which is derived from natural or synthetic sources [4]. Natural products were the most consistently successful source of drug leads [5]. Approximately 60% of drugs in clinical trials against cancer are either natural products or inspired pharmacophores derived from natural products [3,6].

*P. odorata*, popularly known as ‘Alagaw’ is a small tree native to the tropical regions [4]. In the Philippines, the decoction of the leaves is used as a diuretic, a febrifuge, and for the treatment of vaginal irritation, coughs, abdominal pains, and dysentery [5]. Recently, in vitro and in vivo studies targeting its antituberculosis activity showed that *P. odorata* oil along with 1-heneicosyl formate that was isolated from the leaves significantly alleviated tuberculosis [4,6]. *P. odorata* is among the top natural sources of aliphatic iridoids, acylated iridoid glycosides, acylated rhamnopyranosides, flavones, and phenyl ethanoids [5,7,8,9]. Previous investigations of these constituents revealed their inhibitory activity against triple-negative breast cancer cell lines by targeting c-Met phosphorylation [5].

On the other hand; metabolomics is a valuable and comprehensive analysis tool that is used to study the metabolite profiles of unicellular and multicellular biological systems [10]. Plants represent a major challenge in metabolomics due to the high chemical diversity of their metabolites [10]. Consequently, no single analytical method can determine all plant metabolites simultaneously, but LC–HRESIMS is a powerful analytical tool for metabolic profiling that can detect/dereplicate a wide range of chemical compounds at the same time without tedious isolation procedures [11,12].

A limited number of phytochemical and biological investigations have been conducted on the *Premna* species, notably *P. odorata*, where iridoids, flavonoids, phenyl ethanoids, and acylated rhamnopyranosides were described as their major constituents present in leaves [5]. The present study was undertaken to investigate and compare *P. odorata* metabolites presented in different organs (bark, wood, young stems, flowers, and fruits) using LC–HRESIMS, in addition to a phytochemical study targeting the bark organ, which was accompanied by the determination of the cytotoxic activities of the isolated lignans with a pharmacophore study for the active compound.

## 2. Results and Discussion

### 2.1. Metabolomic Analysis 

Chemical profiling of the secondary metabolites of the *P. odorata* bark, wood, young stems, flowers and fruits using LC–HRESIMS for dereplication purposes, resulted in the characterization of a variety of constituents, among which iridoids, flavones, tetrahydrofurofuran lignans, and phenyl ethanoids (Table 1, Figure 1) were predominant, with qualitative and quantitative variation in each organ.

From the metabolomics data, the mass ion peak at *m*/*z* 611.1915 for the predicted molecular formula C_27_H_30_O_16_ was dereplicated as the flavonoid glycoside luteolin-6,8-di-c-*β*-d-glucopyranoside **1**, which was previously detected in *Viola yedoensis* and first reported in the *Premna* genus [13], whereas that at *m*/*z* 387.1276, corresponding to the suggested molecular formula C_21_H_22_O_7_ was dereplicated as 9-hydroxy-3’,4’-dimethoxy-3,4-methylenedioxy-7,9’:7’,9-diepoxylignan **2**, which was formerly reported from the species *Geranium thunbergii* [14]. Likewise, another tetrahydrofurofuran lignan with the molecular formula C_20_H_18_O_8_ was characterized as 4,8-dihydroxy sesamin **3** from the mass ion peak at *m/z* 387.1323, which was previously obtained from *Gmelina arborea* [15]; this was the first report for this metabolite in the genus *Premna*. Moreover, the mass ion peak at *m*/*z* 595.2334, in agreement with the predicted molecular formula C_27_H_30_O_15_ was dereplicated as flavonoid glycoside vicenin-2 **4**. This has been isolated earlier from *Erythrina caffra* [16] but herein its extraction from *P. odorata* plants was reported for the first time. The mass ion peaks at *m*/*z* 581.1560, 551.1592, and 565.2288 for the predicted molecular formulas C_26_H_28_O_15_, C_25_H_26_O_14_, and C_26_H_28_O_14_ were dereplicated as the flavonoid glycoside luteolin 6-c-*β*-d-glucopyranoside, 8-c-*α*-l-arabinopyranoside, luteolin-6,8-di-c-*α*-l-arabinopyranoside, and schaftoside-2, **5**, **6**, **7**, respectively, which were previously detected in *Apometzgeria pubescens* and was first reported in the *Premna* genus [17]. Whereas, the mass ion peaks at *m*/*z* 433.1361, corresponding to the suggested molecular formula C_21_H_20_O_10_, was dereplicated vitexin **8**, which was formerly reported in *P. odorata* [5]. Likewise, acylated iridoid with the molecular formula C_39_H_44_O_20_ was characterized as premnoside-A **9** from the mass ion peak at *m*/*z* 833.2746 and was previously obtained from *P. odorata* [9]. Additionally, a compound at *m*/*z* 535.2889, corresponding to the suggested molecular formula C_25_H_26_O_13_ was dereplicated as flavonoid glycoside apigenin 6,8-di-c-*α*-l-arabinopyranoside **10**, which was formerly reported from the species *Apometzgeria pubescens* and was first reported in the *Premna* genus [17].

In addition to the above-mentioned molecules, the mass ion peaks at *m*/*z* 671.1910 and 671.1910 for the suggested molecular formulas C_30_H_38_O_17_ and C_30_H_38_O_17_ were identified as 6-*o*-*α*-l-(2’’-*O*-*trans*-caffoyl) rhamnopyranosyl catalpol **11** and 6-*o*-*α*-l-(3’’-*O*-*trans*-caffoyl) rhamnopyranosyl catalpol **12**, respectively, which were previously reported in *P. odorata* [8]. Peaks at *m*/*z* 793.3148 and 549.1555 were consistent with the molecular formula C_40_H_34_O_15_ and C_26_H_28_O_13_, and were dereplicated as premnadimer **13** and 4*β*-hydroxyasarinin-1-*o*-*β*-d-glucopyranoside **14**, respectively; tetrahydrofurofuran lignans were previously isolated from the *P. integrifolia* bark [18].

Other iridoids, phenyl ethanoid, and flavones related compounds which were formally isolated from *P. odorata* leaves were characterized as premnoside-B **15**, premnoside-F **16**, premnoside-D **17**, acacetin **18**, premnoside-E **19**, 6- *o*-*α*- l-(4’’-*o-trans*-*P*-methoxycinnamoyl) rhamnopyranosylcatalpol **20**, verbascoside **21**, premnoside-H **22**, premnoside-G **23**, diosmetin **24**, premnaodoroside A-C **25**–**27**, premnoside-C **28**, luteolin **30**, and apigenin **33** based on the mass ion peaks and in accordance with their molecular formulas (Table 1, Figure 1) [5,7,9].

On the other hand, many other tetrahydrofurofuran lignans were also described. In this regard, the mass ion peak at *m*/*z* 355.1937, corresponding to the suggested molecular formula C_20_H_18_O_6_, was dereplicated as sesamin **29**, while that at *m*/*z* 379.2381 was dereplicated as 9-hydroxypinoresinol **31** with the molecular formula C_20_H_22_O_7_ [19,20]. Pinoresinol **32** and 1,5*α*-dihydroxypinoresinol **34** were also characterized by the mass ion peaks at *m*/*z* 359.4261 and 391.2497, in agreement with the predicted formula C_20_H_22_O_6_ and C_20_H_22_O_8_, respectively. The latter four metabolites were reported for the first time in the *Premna* genus (Table 1) [21,22].

It is worth mentioning that based on dereplication, phenyl ethanoids (verbascoside **30**) and flavones (vitexin **8**, acacetin **18**, diosmetin **24**, luteolin **30**, and apigenin **33**) were generally found in all *P. odorata* organs (Table 1). On the other hand; lignans were the major metabolites class in both bark and wood organs, while iridoid metabolites were more associated with the flowers, fruits, and young stems (Table 1, Figure 1).

This is the first report for the active metabolites of *P. odorata* bark; wood, young stems, flowers, and fruits, and these phytochemical data concerning this rarely studied species are of appreciable chemotaxonomic value too.

### 2.2. Phytochemical Investigation of P. odorata Bark

According to LC–HRESIMS metabolomic profiling, iridoids, flavones, and phenyl ethanoids were mainly dominant in the young stems, flowers, and fruits organs, and as previously reported, in leaves. However, bark and wood similarly produced lignans and flavonoid metabolites which were remarkably different from those other organs (Table 1). Consequently, this phytochemical study targeted bark organ polyphenolic compounds.

Based on the physicochemical and chromatographic properties, spectral analyses (UV, ^1^H, and DEPT‒Q NMR), as well as comparison with the literature and some authentic samples, the compounds isolated (Figure 2) from the ethanolic extract of *P. odorata* bark afforded the new tetrahydrofurofurnan lignan **35**, along with the known 4*β*-hydroxyasarinin-1-*o*-*β*-glucopyranoside **36** [18], 1,5*α*-dihydroxypinoresinol **37** [22], premnadimer-A **38** [18], 3-[3′-methoxy-4′-(4′’-formyl-2′’,6′’-dimethoxy-phenoxy)-phenyl]-propenal **39** [23], sinapaldehyde **40** [24], 3,5-dimethoxybenzaldehyde **41** [24], apigenin **42**, luteolin **43**, diosmetin **44**, vitexin **45** [5], orientin **46** [25],vicenin-2 **47** [16], schaftoside-2 **48** [17], and apigenin-6,8-di-c-*α*-l-arabinopyranoside **49** (Figure 2) [25]. All characterized compounds (**37**, **39**–**41**, **46**–**49**) were isolated herein for the first time from the genus *Premna*, while compounds **36**, **38** were isolated for the first time from the *Premna* species.

Compound 35 (Table 2, Figure 2, see Appendix A) was obtained as a white amorphous solid. The HRESIMS data for compound **35** showed an adduct pseudo-molecular ion peak at *m*/*z* 393.0949 [M + Na]^+^, consistent with the molecular formula C_20_H_18_O_7_ and suggesting 12 degrees of unsaturation. The ^1^H, DEPT-Q, and HSQC NMR data (Table 2, Figure 2), showed 6 characteristic resonances; one CH group at *δ*_H_ 2.93 (1H, dd, *J* = 7.3, 7.5) *δ*_C_ 62.1, two benzylic oxymethine moieties at 4.84 (1H, d, *J* = 7.01) *δ*_C_ 83.3 and 4.96 (1H, d, *J* = 10.1) *δ*_C_ 88.1, one oxymethylene group at *δ*_H_ 4.01 (dd, *J* = 9.1, 11.1), 4.22 (dd, *J* = 9.1, 6.1) *δ*_C_ 72.2, one methine at *δ*_H_ 3.91 (1H, dddd, *J* = 6.1, 7.1, 7.3, 11.1) *δ*_C_ 53.2, and one oxymethine at *δ*_H_ 5.57 (1H, s) *δ*_C_ 101.7. Additionally, the HMBC experiment showed the characteristic ^2^*J* HMBC correlation of proton H-1 *δ*_H_ 3.91 (1H, dddd, *J* = 6.1, 7.1, 7.3, 11.1) with C-2 *δ*_C_ 83.3, C5 *δ*_C_ 62.1, C8 *δ*_C_ 72.2. It also showed characteristics of proton H-5 *δ*_H_ 2.93 (1H, ddd, *J* = 7.3, 7.5, 10.1) with C-1 *δ*_C_ 53.2, C-4 *δ*_C_ 101.7, and C-6 *δ*_C_ 88.1, suggesting the characteristic structure of the tetrahydrofurofuran unit [18]. This was confirmed through the ^1^H-^1^H COSY data of 35 (Figure 3), which showed the expected coupling of H-1, H-2, H-4, H-5, H-6, and H-8. Moreover; the acquired NMR data (Table 2, Figure 3) showed that 6 characteristic aromatic resonances appeared at *δ*_H_ 6.78–7.06, where the HMBC experiment showed characteristic ^2^*J* HMBC correlation of proton H-5’ *δ*_H_ 6.81 (1H, d, *J* = 8.0) with C-6’ *δ*_C_ 119.2 and C-4’ *δ*_C_ 147.3, respectively. Additionally, the ^2^*J* HMBC correlation of proton H-5’’ *δ*_H_ 6.89 (1H, d, *J*=8.1) with C-6’’ *δ*_C_ 120.0 and C-4’’ *δ*_C_ 147.2, respectively, suggested the presence of two 1,3,4-trisubstituted benzene rings [18]. The remaining two resonances present in the ^1^H, DEPT-Q, HSQC NMR data perfectly matched to the two OCH_2_O groups that appeared at *δ*_H_ 5.41 (4H, s) *δ*_C_ 101.4. The HMBC experiment showed characteristic ^3^*J* HMBC correlations of the protons of the two OCH_2_O group with C-3’ *δ*_C_ 147.3, C-4’ *δ*_C_ 147.2, C-3’’ *δ*_C_ 147.4, and C-4’’ *δ*_C_ 147.3, respectively (Table 2, Figure 3). The ^2^*J* HMBC correlations of the proton H-2 *δ*_H_ 4.84 (1H, d, *J* = 7.1) *δ*_C_ 83.3, with the quaternary carbonyl carbon C-1’ *δ*_C_ 135.4 and the correlation of the proton H-6 *δ*_H_ 4.96 (1H, d, *J* = 10.1) *δ*_C_ 88.1 with the carbonyl carbon C-1’’ *δ*_C_ 136.1, confirmed the location of the two 1,3,4-trisubstituted benzene rings at carbons C-2 and C-6 at the tetrahydrofurofuran moiety (Figure 3). The relative stereochemistry of compound **35** was deduced using the *J* values and a Nuclear Overhauser Effect (NOE) experiment (Table 2). Where the NOEs observed between H-1, H-2, H-4, and H-5 suggested that they are in the same plan, while the NOEs observed between H-4 and H-6 suggested they are on the other side of the tetrahydrofurofuran moiety. Comparing the ^1^H and DEPT-Q data of compound **35** with the known related compound 4*β*-hydroxyasarinin-1-*o*-*β*-glucopyranoside 36 [18] in which the *o*-*β*-glucopyranoside moiety was attached at C-1, showed that the 1-*o*-*β*-glucopyranoside moiety was obviously absent and the quaternary C-1 at 4*β*-hydroxyasarinin-1-*o*-*β*-glucopyranosid was converted to a methine group that appeared at *δ*_H_ 3.91 (1H, dddd, *J* = 6.1, 7.1, 7.3, 11.1) *δ*_C_ 53.2. Accordingly, compound **35** was identified as 4*β*-hydroxyasarinin.

### 2.3. Cytotoxic Activity

Lignans are a class of secondary plant metabolites that are produced from shikimic acid via the phenylpropanoid pathway through the oxidative dimerization of two phenylpropanoid units [26]. Some lignins proved beneficial in decreasing the risk of breast tumor, prostate cancer, and cardiovascular disease. Further, they can have antioxidant and anti-inflammatory actions [26,27,28]. In this investigation; the isolated compounds **35**–**38** from the ethanolic extract of *P. odorata* bark were evaluated for their cytotoxic activities against HL-60, HT-29, and MCF-7 cancer cell lines, using the MTT assay. The results showed that only compound **35** displayed cytotoxic effect towards HL-60 and MCF-7 cells with IC_50_ values of 2.7 and 4.2 mg/mL, respectively; the known doxirubicin (IC_50_ 4.3, 6.5 µg/mL, respectively) was used as a positive control (Table 3). These results might be associated with the free position at C-1 with the di-substitution of a methylenedioxy group in compound **35** (Figure 2), as compared to the other tested tetrahydrofurofuran lignan compounds **36**–**38**.

Comparing the cytotoxic activity of compound **35** with the related previously isolated isomer (-)-sesamin [28], which has the same carbon framework as compound **35** but differ in the chemical configuration of C-6 and dehydroxylation at C-4, showed that (-)-sesamin exhibited a more potent inhibitory effect on MCF-7 cancer cell invasion with an IC_50_ value of 3.4 µg/mL.

### 2.4. Pharmacophore Model Generation

Pharmacophore modeling is an essential concept in rational drug design, which represents the notion that compounds are active/inactive at a specific receptor because they have/lack some crucial functional groups (features) that correspondingly interact with specific moieties in the binding site of the target receptor [29,30]. Pharmacophore modeling has empowered the rational design of more active ligands, the discovery of more active hits by screening molecular databases, the generation of 3D SAR, the design of hits without the necessity of a receptor structure, and mapping the chemical space occupied by active ligands [29,30].

The HipHop module of the Discovery Studio 2.5 software was applied to construct a pharmacophore hypothesis of compounds **35**–**38** (Figure 4). The highest-ranked model (generated using parameters described in the experimental part) had the following features which mapped compound **35**—two hydrogen bond acceptor (HBAs) aligning the phenoxy oxygen atoms of the benzodioxole moieties, two Ring Arom features vectored on the two phenyl rings, one hydrogen bond donor (HBD) feature aligning the central hydroxyl group, and thirteen exclusion spheres limit the boundaries of the sterically inaccessible regions of the target’s active site.

The advantage of this model is that it identifies the essential chemical features necessary to inhibit the growth of cancer cells. Moreover, the generated pharmacophore can map the boundaries of the chemical space by placing exclusion constrains that mimic the inaccessible region of the active site.

## 3. Materials and Methods

### 3.1. Plant Material

*P. odorata* bark, wood, young stems, flowers, and fruits were collected in May 2018 from the Giza Zoo garden, Egypt. *P. odorata* was kindly identified by Dr. Abd ElHalim A. Mohammed of the Horticultural Research Institute, Department of Flora and Phytotaxonomy Researches, Dokki, Cairo, Egypt. A voucher specimen (2018-BuPD 45) was deposited at the Department of Pharmacognosy, Faculty of Pharmacy, Beni-Suef University, Egypt.

### 3.2. Chemicals and Reagents

Solvents used in this work, such as n-hexane (boiling point b.p. 60–80 °C), dichloromethane (CH_2_Cl_2_), ethyl acetate (EtOAc), ethanol (EtOH), and methanol (MeOH), were purchased from the El-Nasr Company for Pharmaceuticals and Chemicals, Egypt; the solvents were distilled before use. Deuterated solvents (Sigma-Aldrich, Saint Louis, Missouri, USA), including chloroform (CDCl_3_), methanol (CD_3_OD), and dimethyl sulfoxide (DMSO-*d_6_*) were used for the nuclear magnetic resonance (NMR) spectroscopic analyses. On the other hand, column chromatography (CC) was performed using silica gel 60 (63–200 μm, E. Merck, Sigma-Aldrich), polyamide-6 (50–160μm), and Sephadex LH–20 (0.25–0.1 mm, GE Healthcare, Sigma-Aldrich), while silica gel GF_254_ for Thin-layer chromatography (TLC) (El-Nasr Company for Pharmaceuticals and Chemicals, Egypt) was employed for vacuum liquid chromatography (VLC). Thin-layer chromatography (TLC) was carried out using pre-coated silica gel 60 GF_254_ plates (E. Merck, Darmstadt, Germany; 20 × 20 cm, 0.25 mm in thickness). Spots were visualized by spraying with para-anisaldehyde reagent PAA (85:5:10:0.5 absolute ethanol:sulfuric acid:glacial acetic acid:para-anisaldehyde), followed by heating at 110 °C [31]. Aluminum chloride reagent (5% in ethanol) was also used for the detection of flavonoids on TLC [32]. For the biological study, doxorubicin was used as a positive control and was purchased from the Merck Company, Sigma-Aldrich, Germany, while the HL-60, HT-29, and MCF-7 cancer cell lines were obtained from the American Type Culture Collection (ATCC, Rockville, MD; HPACC, Salisbury, UK) and were routinely sub-cultured twice per week. All other chemicals and reagents were purchased from Sigma Aldrich.

### 3.3. Apparatus

The ^1^H, DEPT-Q, and 2D NMR spectra were recorded at 400 and 100 MHz, respectively; using tetramethylsilane (TMS) as the internal standard in methanol-*d_4_*, using the residual solvent peak (*δ*_H_ = 3.34 & *δ*_C_ = 49.9) as references, on Bruker Avance III 400 MHz with a BBFO Smart Probe and Bruker 400 MHz EON Nitrogen-Free Magnet (Bruker AG, Billerica, MA, USA). Carbon multiplicities were determined using the DEPT-Q experiment. The optical rotation in methanol was obtained using a Perkin-Elmer 343 polarimeter (PerkinElmer Inc., Waltham, MA, USA). The UV spectrum in methanol was obtained using a Shimadzu UV 2401PC spectrophotometer (Shimadzu Corporation - UV-2401PC/UV-2501PC, Kyoto, Japan). The IR spectra were obtained using a Jasco FTIR 300E infrared spectrophotometer. HRESIMS data were obtained using an Acquity Ultra Performance Liquid Chromatography system coupled to a Synapt G2 HDMS quadrupole time-of-flight hybrid mass spectrometer (Waters, Milford, MA, USA).

### 3.4. Extraction and Fractionation of Plant Material

*P. odorata* bark (2 kg) was collected and air-dried in shade for one month. After drying, the bark was finely powdered using a OC-60B/60B set small, corn, bark grinding machine for herbs (60–120 mesh, Henan, Mainland China). The finely powdered bark was exhaustively extracted by maceration with 70% ethanol (5 L, 3 ×, seven days each) at room temperature and concentrated under vacuum at 45 °C, using rotavap (Buchi Rotavapor R-300, Cole-Parmer, Vernon Hills, IL, USA), to afford an 80 g crude extract. The dry extract was suspended in 200 mL distilled water and successively portioned with solvents of different polarities (n-hexane, CH_2_Cl_2_, and EtOAc). The organic phase in each step was separately evaporated under reduced pressure to afford the corresponding fractions I (5.0 g), II (10.0 g), and III (10.0 g), respectively, while the remaining mother liquor was then concentrated to give the aqueous fraction (IV). All resulting fractions were kept at 4 °C for the biological and phytochemical investigations.

### 3.5. Metabolomic Analysis Procedure

Air-dried and finely powdered *P. odorata* bark, wood, young stems, flowers, and fruits (2 g each) were exhaustively extracted with 70% ethanol (5 ml, 3 ×, 0.5 h.) at room temperature and concentrated under vacuum to afford 50, 80, 80, 40, and 50 mg crude extracts, respectively. The crude extracts of different organs were subjected to metabolomic analysis using analytical techniques of LC–HRESIMS, according to Abdelmohsen et al. [33]. LC–HRESIMS metabolomics analyses were performed on an Acquity Ultra Performance Liquid Chromatography system coupled to a Synapt G2 HDMS quadrupole time-of-flight hybrid mass spectrometer (Waters, Milford, MA, USA). Chromatographic separation was carried out on a BEH C_18_ column (2.1 × 100 mm, 1.7 μm particle size; Waters, Milford, MA, USA) with a guard column (2.1 × 5 mm, 1.7μm particle size) and a linear binary solvent gradient of 0–100% eluent B, over 6 min, at a flow rate of 0.3 mL min^−1^, using 0.1% formic acid in water (*v/v*) as solvent A and acetonitrile as solvent B. The injection volume was 2 μL and the column temperature was 40 °C. After chromatographic separation, the metabolites were detected by mass spectrometry using electrospray ionization (ESI) in the positive mode; the source was operated at 120 °C. The ESI capillary voltage was set to 0.8 kV, the sampling cone voltage was set to 25 V and nitrogen (at 350 °C, a flow rate of 800 L h^−1^) was used as the desolvation gas and the cone gas (flow rate of 30 L h^−1^). The mass range for TOF‒MS was set from *m*/*z* (mass-to-charge ratio) 50‒1200. In MZmine 2.12, the raw data were imported by selecting the ProteoWizard converted positive files in the mzML format. Mass ion peaks were detected and followed by a chromatogram builder and a chromatogram deconvolution. The local minimum search algorithm was applied and isotopes were also identified via the isotopic peaks grouper. Missing peaks were detected using the gap-filling peak finder. An adduct search, as well as complex search was performed. The processed data set was then subjected to molecular formula prediction and peak identification. The positive and negative ionization mode data sets from each of the respective plant extract were dereplicated against the DNP database Dictionary of Natural Products (DNP) database.

### 3.6. Isolation and Purification of Compounds

A part of fraction II (8 g) was subjected to VLC fractionation on a silica gel column (6 × 30 cm, 50 g). Elution was performed using CH_2_Cl_2_–EtOAc gradient mixtures in the order of increasing polarities (0%, 5%, 10%, 15%, 20%, 25%, 30%, 35%, 40%, 45%, 50%, 60%, 80%, and 100%, 1L each, flow rate FR 3 mL/min), then with EtOAc–MeOH (50, 50, 1L, flow rate FR 3 mL/min), and finally with MeOH. The effluents were collected in fractions (100 mL each); each fraction was concentrated and monitored by TLC, using the system CH_2_CL_2_:EtOAc in the ratio 9:1 and the PAA reagent. Similar fractions were grouped and concentrated under reduced pressure, to provide three subfractions (II_1_–II_3_). Subfraction II_1_ (1.0 g) was further fractionated on a silica gel column (65 × 1.5 cm, 50 g). Elution was performed using n-hexane–EtOAc gradient mixtures, in order of increasing polarities (0%, 5%, 10%, 15%, 20%, 25%, 30%, 35%, 40%, 45%, 50%, 60%, 80%, and 100%, 250 mL each, FR 3 mL/min). The effluents were collected in fractions (20 mL each); each fraction was concentrated and monitored by TLC to afford compound **35** (7 mg), **38** (50 mg), **40** (6 mg), and **41** (7 mg). While Subfraction II_2_ (1 g) was further fractionated on a silica gel column (65 × 1.5 cm, 50 g). Elution was performed using CH_2_CL_2_–EtOAc gradient mixtures in order of increasing polarities (0%, 5%, 10%, 15%, 20%, 25%, 30%, 35%, 40%, 45%, 50%, 60%, 80%, and 100%, 250 mL each, FR 3 mL/min). The effluents were collected in fractions (20 mL each); each fraction was concentrated and monitored by TLC to afford compound **37** (15 mg). Finally, Subfraction II_3_ (0.5 g) was further fractionated on a silica gel column (100 × 1 cm, 25 g). Elution was performed using a CH_2_CL_2_–EtOAc isocratic mixture (1%, 500 mL, FR 3 mL/min) to afford compound **39** (50 mg).

Moreover, a part of EtOAc fraction III (8 g) was fractionated on a polyamide column (50–160 μm, 1000 × 5 cm, 100 g) using a gradient elution starting with H_2_O and ending with MeOH in the order of increasing polarities (0%, 5%, 10%, 15%, 20%, 25%, 30%, 35%, 40%, 45%, 50%, 60%, 80%, and 100%, 1000 mL each, FR 5 mL/min). The effluents were collected in fractions (200 mL each); each fraction was concentrated and monitored by TLC using the system EtOAc:glacial acetic acid formic acid:H_2_O in the ratio 10:1:1:2 and the PAA reagent. Similar fractions were grouped together and concentrated under reduced pressure to also provide three subfractions (III_1_–III_3_). Subfraction III_1_ (1.0 g) was further fractionated on a silica gel column (63–200 μm, 65 × 1.5 cm, 50 g). Elution was performed using CH_2_Cl_2_–MeOH gradient mixtures, in order of increasing polarities (0%, 5%, 10%, 15%, 20%, 25%, 30%, 35%, 40%, 45%, 50%, 60%, 80%, and 100%, 250 mL each, FR 3 mL/min). The effluents were collected in fractions (20 mL each); each fraction was concentrated and monitored by TLC using the system EtOAc:glacial acetic acid:formic acid:H_2_O in the ratio 10:1:1:2 and PAA reagent, to afford one promising sub-subfraction III_1_ (80 mg), which was further purified on a Sephadex LH_20_ column (0.25–0.1 mm, 100 × 0.5 cm, 100 gm) which eluted with MeOH to afford compound **36** (17 mg). While Subfraction III_2_ (1 g) was further fractionated on a silica gel column (63–200 μm, 65 × 1.5 cm, 50 g). Elution was performed using CH_2_Cl_2_–MeOH gradient mixtures in order of increasing polarities (0%, 5%, 10%, 15%, 20%, 25%, 30%, 35%, 40%, 45%, 50%, 60%, 80%, and 100%, 250 mL each, FR 3 mL/min). The effluents were collected in fractions (20 mL each); each fraction was concentrated and monitored by TLC using the system EtOAc:glacial acetic acid:formic acid:H_2_O in the ratio of 10:1:1:2 and the PAA reagent, to afford one promising sub-subfraction III_2_ (70 mg), which further subjected to a preparative TLC eluted with the system EtOAc: glacial acetic acid: formic acid: H_2_O 10:1:1:2 to give compounds **47** (10 g), **48** (13 mg), and **49** (17 mg). Finally, Subfraction III_3_ (1.5 g) was further fractionated on a silica gel column (63–200 μm, 65 × 1.5 cm,75 g). Elution was performed using CH_2_Cl_2_-MeOH gradient mixtures in order of increasing polarities (0, 1, 2, 3, 4, 5, 6, 7, 8, 9, 10, 15, 20, 25, 30, 35, 40, 45, 50, 60, 80 and 100%, 500 mL each, FR 3 mL/min). The effluents were collected in fractions (20 mL each); each fraction was concentrated and monitored by TLC to afford five promising sub-subfraction III_3_, each sub-subfraction III_3_ was further separately purified on a Sephadex LH_20_ column (0.25–0.1 mm, 100 × 0.5 cm, 100 g) which eluted with MeOH to afford compounds **42** (10 gm), **43** (13 mg), **44** (20 mg), **45** (20 mg), and **46** (17 mg).

**4*β*-hydroxyasarinin** (**35**): Pale white amorphous solid; mp: 100–102 °C (lit.^19^ 99–101 °C); TLC (CH_2_Cl_2_:EtOAC, 90:10 *v/v*): R_f_ = 0.20; ${\left[\alpha \right]}_{D}^{25}$ +18.5 (c 0.15, MeOH); UV (MeOH) λ_max_ (log _ε_) 206 (1.4), 238 (1.6), 285 (0.8) nm; IR υ_max_ (KBr) 3639, 3100, 1600, 1530, 1420, 1250, 1200, 1100, 853, 703, 601 cm^−1^; NMR data: refer to Table 1; HRESIMS *m*/*z* 393.0949 [M +Na]^+^ (calc. for C_20_H_18_NaO_7_, 393.0945).

### 3.7. Cell Cultures

HL-60 cells (human promyelocytic cell) were grown in 5% (*v/v*) CO_2_ in an RPMI 1640 medium at 37 °C, supplemented with 10% (*v/v*) 1%(*w/v*) l-glutamine, and 0.4% (*w/v*) antibiotics (50 U/mL penicillin and 50 mg/mL streptomycin). The HT-29 cells (human colon adenocarcinoma cell) were cultured at 37 °C, 5% (*v/v*) CO_2_ in Dulbecco’s Modified Eagle Medium (DMEM) with high glucose (4.5 g/L), supplemented with 10% (*v/v*) fetal bovine serum (FBS), 1% (*w/v*) l-glutamine, and 0.4% (*w/v*) antibiotics (50 U/mL penicillin, 50 mg/mL streptomycin). MCF-7 cells (Human breast adenocarcinoma cell) were cultured at 37 °C, 5% (*v/v*) CO_2_ in an RPMI1640 medium, supplemented with 10% (*v/v*) fetal bovine serum (FBS), 1% (*w/v*) L-glutamine, 1% sodium pyruvate, and 0.4% (*w/v*) antibiotics (50 U/mL penicillin, 50 mg/mL streptomycin).

### 3.8. Cytotoxic Assessment 

The cytotoxic properties of compounds **35**–**38** were evaluated against the three cancer cell lines including the HL-60, HT-29, and MCF-7 cell lines [34]. Cell proliferation was evaluated in the cell lines by the MTT assay, in triplicates. In brief, the cells were placed in a 96-well microtiter plate at a density of 1 × 10^4^ cells per well, in a final volume of 100 μL of culture medium. These cells were treated for 24 h with isolated compounds, using 10 µL MTT (5.0 mg/mL) at 37 °C with 5% CO_2_. After treatment, the cells were immediately incubated for 4 h at 37 °C. The cells were lysed in 100 μL of lysis buffer (isopropanol, conc. HCl and Triton X-100) for 10 min at room temperature and 300 rpm/min. The enzymatic reduction of MTT to formazan crystals that dissolved in DMSO was quantified by a Varioskan LUX multimode reader (Bio Tek, Tokyo, Japan) at 570 nm. Dose-response curves were generated and the IC_50_ values were defined as the concentration of compound required to inhibit cell proliferation by 50%. Doxirubicin was used as a positive control.

Data were expressed as mean ± S.E.M (*n* = 3). One-way analysis of variance (ANOVA) followed by Dunnett’s test was applied. Graph Pad Prism 5 was used for statistical calculations (Graph pad Software, San Diego, California, USA). Results were regarded as significant at *p* ˂ 0.05.

### 3.9. Pharmacophore Model Generation

ChemBioDraw Ultra 12.0 (CambridgeSoft, 100 Cambridge Park Drive, Cambridge, MA 02140) was used to sketch the 2D chemical structures of compounds **35**–**38**, saved in the MDL-molfile format and were subsequently imported into the Discovery Studio 2.5 to be converted into corresponding 3D structures [29,30]. The HipHop module of the Discovery Studio 2.5 software was applied to construct plausible binding hypotheses for compounds **35**–**38**. HipHop identifies 3D spatial arrangements of chemical features that are common to active molecules. HipHop was instructed to explore five to six-featured pharmacophoric spaces of the following possible features; hydrogen bond donor (HBD), hydrogen bond acceptor (HBA), hydrophobic (Hbic), and ring aromatic (Ring Arom). The number of features was allowed to vary from 1–2 for HBD, 1–3 for HBA, 0–3 for Hbic, and 0–3 for Ring Arom. HipHop uses inactive compounds to create excluded volumes that resemble the sterically inaccessible regions of the binding pocket. HipHop was configured to allow a maximum of 20 exclusion spheres to be added to the generated pharmacophoric hypotheses. The user defines the degree of molecules mapping completely or partially to the hypothesis, via the Principal and MaxOmitFeat parameters of the software. The conformational space of each compound was explored applying the ‘‘best conformer generation’’ option. The default values were applied for other parameters.

## 4. Conclusions

This study dealt with LC–HRESIMS metabolomic profiling of *P. odorata* different organs (bark, wood, young stems, flowers, and fruits) which derplicated different chemical metabolomic compounds belonging to different classes, including iridoids, flavones, phenyl ethanoids, and lignans. A phytochemical investigation of the *P. odorata* bark afforded one new tetrahydrofurofuran lignan, 4*β*-hydroxyasarinin 35, along with fourteen known compounds. Investigating the cytotoxic activities of compounds **35**–**38** against the HL-60, HT-29, and MCF-7 cancer cell lines, using the MTT assay showed that only compound **35** had cytotoxic effects against HL-60 and the MCF-7 cell lines, with IC_50_ values of 2.7 and 4.2 mg/mL, respectively, using a concentration 10 µL MTT (5.0 mg/mL). The HipHop module of 4*β*-hydroxyasarinin showed two HBAs aligning with the phenoxy oxygen atoms of benzodioxole moieties, two Ring Arom features vectored on the two phenyl rings, one HBD feature aligning the central hydroxyl group, and thirteen exclusion spheres limiting the boundaries of the sterically inaccessible regions of the target’s active site.

## Figures and Tables

**Figure 1 metabolites-09-00223-f001:**
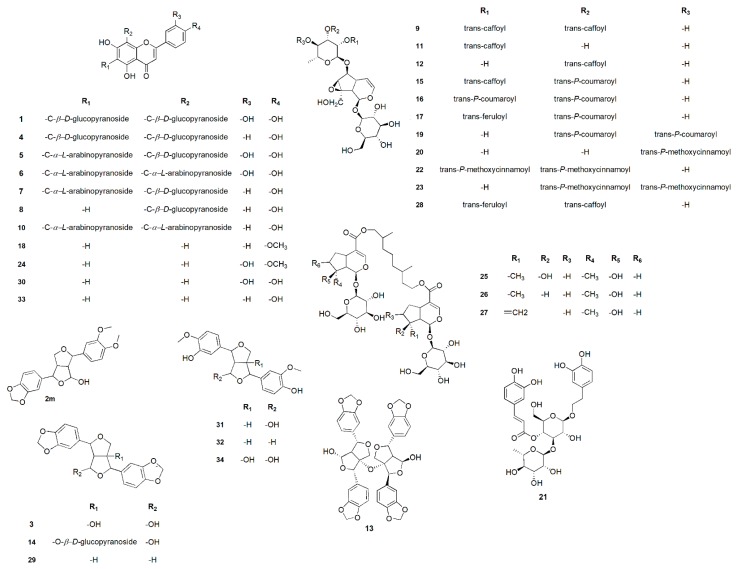
Dereplicated metabolites from the LC–HRESIMS analysis of *Premna odorata* bark, wood, young stems, flowers, and fruit.

**Figure 2 metabolites-09-00223-f002:**
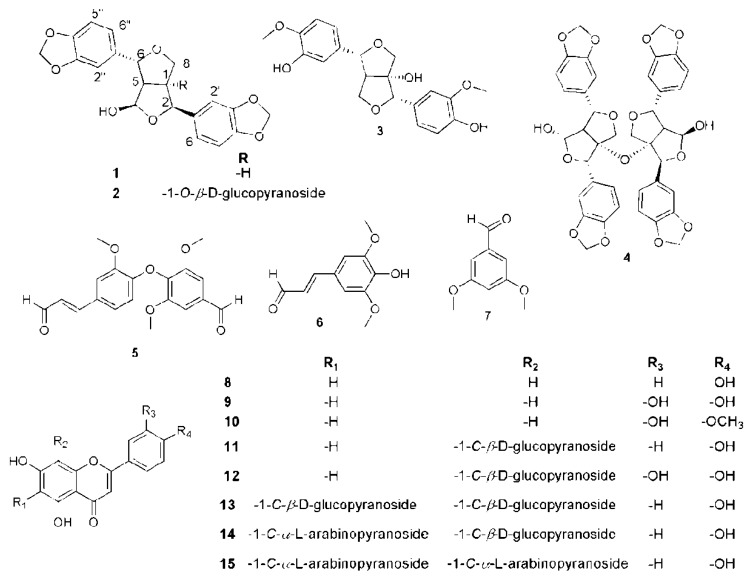
Structures of compounds isolated from the *Premna odorata* bark.

**Figure 3 metabolites-09-00223-f003:**
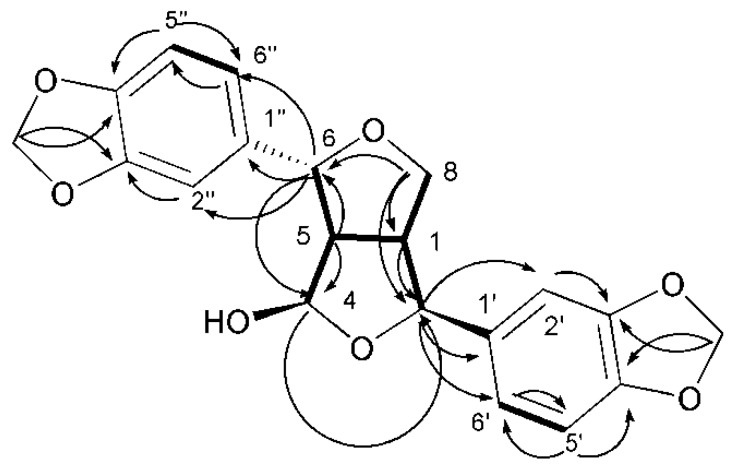
Selected **HMBC** (
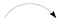
) and ^1^H–^1^H COSY (
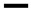
) correlations of compound **35**.

**Figure 4 metabolites-09-00223-f004:**
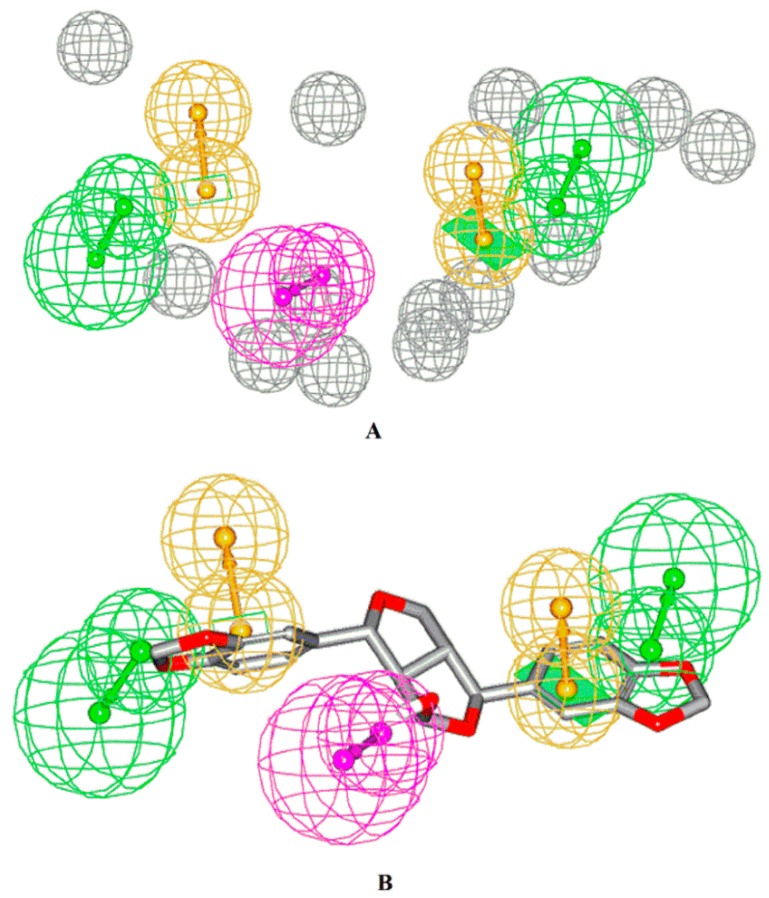
(**A**) The highest-ranked pharmacophore model generated for compounds **35**–**38**. Hydrogen bond acceptor (HBAs) are represented as green-vectored spheres, hydrogen bond donor (HBD) is represented as a purple-vectored sphere, Ring Arom features are represented as an orange-vectored sphere, and the exclusion volumes as gray spheres. (**B**) Structure of compound **35** and how it aligns the pharmacophore map. Exclusion spheres were hidden for clarity.

**Table 1 metabolites-09-00223-t001:** The LC–HR–ESIMS dereplication results of the *Premna odorata* bark, wood, young stems, flowers, and fruits crude extracts.

No.	Metabolites Name	Original Source	MF	RT(min.)	*m*/*z*	B.	W.	Y.	Fl.	Fr.
**1**	Luteolin-6,8-di-c-*β*-d-glucopyranoside	*Viola yedoensis*	C_27_H_30_O_16_	7.54	611.1915	+	+			
**2**	9-hydroxy-3’,4’-dimethoxy-3,4-methylenedioxy-7,9’:7’,9-diepoxylignan	*Geranium thunbergii*	C_21_H_22_O_7_	7.60	387.1276	+	+			
**3**	4,8-dihydroxy sesamin	*Gmelina arborea*	C_20_H_18_O_8_	7.71	387.1323	+	+			
**4**	Vicenin-2	*Erythrina caffra*	C_27_H_30_O_15_	8.25	595.2334	+	+			
**5**	luteolin 6-c-*β*-d-glucopyranoside, 8-c-*α*-l-arabinopyranoside	*Apometzgeria pubescens*	C_26_H_28_O_15_	8.65	581.1560	+	+			
**6**	Luteolin-6,8-di-c-*α*-l-arabinopyranoside	*Apometzgeria pubescens*	C_25_H_26_O_14_	8.89	551.1592	+	+			
**7**	Schaftoside-2	*Apometzgeria pubescens*	C_26_H_28_O_14_	9.11	565.2288	+	+			
**8**	Vitexin	*Premna odorata*	C_21_H_20_O_10_	9.71	433.1361	+	+	+	+	+
**9**	Premnoside A	*Premna odorata*	C_39_H_44_O_20_	11.37	833.2746				+	+
**10**	Apigenin-6,8-di-c-*α*-l-arabinopyranoside	*Apometzgeria pubescens*	C_25_H_26_O_13_	11.81	535.2889	+	+			
**11**	6- *o*-*α*- l-(2″-*o*-*trans*-caffoyl) rhamnopyranosylcatalpol	*Premna odorata*	C_30_H_38_O_17_	11.98	671.1910				+	+
**12**	6- *o*-*α*- l-(3″-*o*-*trans*-caffoyl) rhamnopyranosylcatalpol	*Premna odorata*	C_30_H_38_O_17_	12.03	671.1910				+	+
**13**	Premnadimer A	*Premna integrifolia*	C_40_H_34_O_15_	12.22	793.3148	+	+			
**14**	4*β*-hydroxyasarinin-1-*o*-*β*-d-glucopyranoside	*Premna integrifolia*	C_26_H_28_O_13_	12.53	549.1555	+	+			
**15**	Premnoside B	*Premna odorata*	C_39_H_44_O_19_	12.61	817.2282				+	+
**16**	Premnoside F	*Premna odorata*	C_39_H_44_O_18_	13.00	801.2404			+	+	+
**17**	Premnoside D	*Premna odorata*	C_40_H_46_O_19_	13.09	831.2411				+	+
**18**	Acacetin	*Premna odorata*	C_16_H_12_O_5_	13.60	285.1126	+	+	+	+	+
**19**	Premnoside E	*Premna odorata*	C_39_H_44_O_18_	13.72	801.2404			+	+	+
**20**	6- *o*-*α*- l-(4’’-*o-trans*-*p*-methoxycinnamoyl) rhamnopyranosylcatalpol	*Premna odorata*	C_31_H_40_O_16_	14.10	669.1634				+	+
**21**	Verbascoside	*Premna odorata*	C_29_H_36_O_15_	14.47	625.1396	+	+	+	+	+
**22**	Premnoside H	*Premna odorata*	C_41_H_48_O_18_	14.54	829.2011			+	+	+
**23**	Premnoside G	*Premna odorata*	C_41_H_48_O_18_	14.73	829.2011			+	+	+
**24**	Diosmetin	*Premna odorata*	C_16_H_12_O_6_	15.09	301.2947	+	+	+	+	+
**25**	Premnaodoroside A	*Premna odorata*	C_42_H_66_O_20_	15.94	891.3561				+	+
**26**	Premnaodoroside B	*Premna odorata*	C_42_H_66_O_19_	16.06	875.2437				+	+
**27**	Premnaodoroside C	*Premna odorata*	C_42_H_64_O_19_	16.12	873.3192				+	+
**28**	Premnoside C	*Premna odorata*	C_40_H_46_O_20_	16.21	847.2782				+	+
**29**	Sesamin	*Zanthoxylum senegalense*	C_20_H_18_O_6_	18.21	355.1937	+	+			
**30**	Luteolin	*Premna odorata*	C_15_H_10_O_6_	18.66	309.2349	+	+	+	+	+
**31**	9-hydroxypinoresinol	*Allamanda neriifolia*	C_20_H_22_O_7_	19.07	379.2381	+	+			
**32**	Pinoresinol	*Festuca argentina*	C_20_H_22_O_6_	20.61	359.4261	+	+			
**33**	Apigenin	*Premna odorata*	C_15_H_10_O_5_	20.82	293.2147	+	+	+	+	+
**34**	1,5*α*-dihydroxypinoresinol	*Helianthemum sessiliflorum*	C_20_H_22_O_8_	22.25	391.2497	+	+			

MF: molecular formula, RT: retention time, min: minute, B: bark, W: wood, Y: young stems, Fl: flowers, Fr: fruits.

**Table 2 metabolites-09-00223-t002:** DEPT-Q (400 MHz) and ^1^H (100 MHz) NMR data of compound **35** in CDCL_3_; Carbon multiplicities were determined by the DEPT-Q experiments.

Position	*δ* _C_	*^δ^*_H_ (*J* in Hz)
**1**	53.2,CH	3.91 (dddd, 6.1, 7.1, 7.3, 11.1)
**2**	83.3,CH	4.84 (d,7.1)
**3**		
**4**	101.7,CH	5.57 (s)
**5**	62.1,CH	2.93 (dd, 7.3, 7.5)
**6**	88.1,CH	4.96 (d, 10.1)
**7**		
**8**	72.2,CH_2_	4.01 (dd, 9.1, 11.1), 4.22 (dd,9.1, 6.1)
**1’**	135.4,C	
**2’**	107.1,CH	7.06 (d,2.1)
**3’**	147.3,C	
**4’**	147.2,C	
**5’**	108.1,CH	6.81 (d,8.0)
**6’**	119.2,CH	6.80 (dd,2.1,8.0)
**OCH_2_**	101.4	5.41 (s)
**1’’**	136.1,C	
**2’’**	106.3,CH	6.86 (d,2.0)
**3’’**	147.4,C	
**4’’**	147.3,C	
**5’’**	108.2,CH	6.89 (d,8.1)
**6’’**	120.0,CH	6.78 (dd,2.0,8.1)
**OCH_2_**	101.4	5.41 (s)

qC—quaternary, CH—methine, CH_2_—methylene, and CH_3_—methyl carbon.

**Table 3 metabolites-09-00223-t003:** Cytotoxic properties for compounds **35**–**38** isolated from *Premna odorata* bark.

Compound	IC_50_ (µg/mL) ^a^		
	HL-60	MCF-7	HT-29
**35**	2.7 ^*^	4.2 ^*^	NA
**36**	NA	NA	NA
**37**	NA	NA	NA
**38**	NA	NA	NA
**Doxirubicin ^b^**	0.08	0.68	501

NA—not active ^a^ IC_50_ value of compounds against each cancer cell line, which was defined as the concentration (µg) that caused a 50% inhibition of cell growth in vitro. Data were expressed as mean ± S.E.M (*n* = 3). One-way analysis of variance (ANOVA) followed by Dunnett’s test was applied. Graph Pad Prism 5 was used for statistical calculations (Graph pad Software, San Diego, California, USA). ^*^ statistically significant at *p* < 0.05. ^b^ Doxirubicin a positive control.

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
