# Peer review of "Metabolomic Profiling and Cytotoxic Tetrahydrofurofuran Lignans Investigations from Premna odorata Blanco"

_metabolites, 2019, doi:10.3390/metabo9100223_

Round 1

Reviewer 1 Report

The Manuscript is well structured in its present form. The introduction is very complete. The experiments are well designed and the results obtained are very interesting. Have been identified different compounds. These compounds have been assessed as antitumoral agents, with very interesting results.

The discussion of the results is very clear, as well as the conclusions obtained. The tables and figures are appropriate to the discussion.

Author Response

Reviewer 1

The Manuscript is well structured in its present form. The introduction is very complete. The experiments are well designed and the results obtained are very interesting. Have been identified different compounds. These compounds have been assessed as antitumoral agents, with very interesting results. The discussion of the results is very clear, as well as the conclusions obtained. The tables and figures are appropriate to the discussion.

Thanks for the nice comments.

Reviewer 2 Report

  For the authors:
the manuscript titled "Metabolomic Profiling and Cytotoxic Tetrahydrofurofuran Lignans Investigations from Premna odorata Blanco" deals with a chemical characterization of several dereplicated compounds in different organs of Premna odorata Blanco. Title and abstract are not persuasive. Many sentences in the text are incomplete, with a large number of careless errors like poor grammar and spelling mistakes. Many times the verbs are also missing making the manuscript difficult to read and assess the quality of the content. A thorough review by a native English speaker is absolutely required before the manuscript can be considered for publication in whatever journal.

Author Response

Reviewer 2

Manuscript by Elmaidomy et al. describes a comprehensive metabolic profiling of plant extracts from Premna odorata by LC-HRESIMS with identification and classification of various chemicals such as iridoids, flavones, phenyl ethanoids, and lignans. Further phytochemical analysis led to the study of a cytotoxic compound, a new tetrahydrofurofuran lignan (compound 35) against cancer cell lines and reported their IC50 values. They also showed the pharmacophore map of compound 35 to inform more potential for further drug design. While the compound 35 is tested to be cytotoxic and pharmacophore map was done, the ability of this compound to bind certain protein target still needs to be verified. In-silico analysis of what metabolic pathways are targeted in cancer cells, such as molecular docking will be more useful for further studies of drug design.

Note: There are some typos that need to be corrected in this manuscript.

Thanks for the nice comments.  The manuscript was undergoing extensive English editing, and grammar checked (highlight yellow).

Molecular docking is a very good tool to clarify the interaction between the compound and the target but in vitro experiment is a demand to validate the in silico suggestion

In our case, there are many targets clarify the activity but it is hard to predict unless we carry out a lot of biological activity including free cell lysate and Western plot which are not available for us now

For that reason, pharmacophore nap plays a great role in describing the important dimensions and functionality of the new compound which was the only active 

Reviewer 3 Report

Manuscript by Elmaidomy et al. describes a comprehensive metabolic profiling of plant extracts from Premna odorata by LC-HRESIMS with identification and classification of various chemicals such as iridoids, flavones, phenyl ethanoids, and lignans. Further phytochemical analysis led to the study of a cytotoxic compound, a new tetrahydrofurofuran lignan (compound 35) against cancer cell lines and reported their IC50 values. They also showed the pharmacophore map of compound 35 to inform more potential for further drug design. While the compound 35 is tested to be cytotoxic and pharmacophore map was done, the ability of this compound to bind certain protein target still needs to be verified. In-silico analysis of what metabolic pathways are targeted in cancer cells, such as molecular docking will be more useful for further studies of drug design.

Note: There are some typos that need to be corrected in this manuscript.

Author Response

Reviewer 3

Only MS1 data was listed in Table 1, was tandem MS performed to increase the confidence level of identification results?

Thanks for the nice comment; but unfortunately we couldn’t carry out the tandem MS due to lack of facilities

The metabolome coverage is very important in untargeted metabolomics. However, only 34 identified metabolites were listed in Table 1. Did the authors detect any other metabolites? If so, how many?

Thanks for the nice comment; almost 106 metabolites had been identified but their reflection in data bases not found.  

In the experimental session, line 298, “After chromatographic separation, metabolites were detected by mass spectrometry using electrospray ionization (ESI) in the positive mode”, however, in the later text, line 307, “The positive and negative ionization mode data sets from each of the respective”. If only the positive mode was used in the LC-MS analysis, why did author have negative mode data set?

Thanks for the nice comment; we correct this sentence

The chromatographic condition is not clear in this manuscript, please clarify.

Thanks for the nice comment; we carefully revised the chromatographic methods in the material and method part.

Other than reversed phase chromatography, HILIC is also a very common separation technique used in metabolomics, did author try HILIC to separate the plant extraction?

Thanks for the nice comment; but unfortunately we didn’t try it.

Line 362 “supplemented with 10% (v/v) 1%(w/v)” should be “10% FBS” 

 Thanks for the nice comment; the correction had been made as requested.

Reviewer 4 Report

In this study, the authors applied LC-MS to profile the metabolome of P. odorata. NMR was performed to confirm the structures of several unknown metabolites. MTT assay was used to evaluate the cytotoxicity of some metabolites extracted from P.odorata. Some issues need to be addressed before considering for publication.

Only MS1 data was listed in Table 1, was tandem MS performed to increase the confidence level of identification results? The metabolome coverage is very important in untargeted metabolomics. However, only 34 identified metabolites were listed in Table 1. Did the authors detect any other metabolites? If so, how many? In the experimental session, line 298, “After chromatographic separation, metabolites were detected by mass spectrometry using electrospray ionization (ESI) in the positive mode”, however, in the later text, line 307, “The positive and negative ionization mode data sets from each of the respective”. If only the positive mode was used in the LC-MS analysis, why did author have negative mode data set? The chromatographic condition is not clear in this manuscript, please clarify. Other than reversed phase chromatography, HILIC is also a very common separation technique used in metabolomics, did author try HILIC to separate the plant extraction? Line 362 “supplemented with 10% (v/v) 1%(w/v)” should be “10% FBS”

Author Response

Reviewer 4

Only MS1 data was listed in Table 1, was tandem MS performed to increase the confidence level of identification results?

Thanks for the nice comment; but unfortunately we couldn’t carry out the tandem MS due to lack of facilities

The metabolome coverage is very important in untargeted metabolomics. However, only 34 identified metabolites were listed in Table 1. Did the authors detect any other metabolites? If so, how many?

Thanks for the nice comment; almost 106 metabolites had been identified but their reflection in data bases not found.  

In the experimental session, line 298, “After chromatographic separation, metabolites were detected by mass spectrometry using electrospray ionization (ESI) in the positive mode”, however, in the later text, line 307, “The positive and negative ionization mode data sets from each of the respective”. If only the positive mode was used in the LC-MS analysis, why did author have negative mode data set?

Thanks for the nice comment; we correct this sentence

The chromatographic condition is not clear in this manuscript, please clarify.

Thanks for the nice comment; we carefully revised the chromatographic methods in the material and method part.

Other than reversed phase chromatography, HILIC is also a very common separation technique used in metabolomics, did author try HILIC to separate the plant extraction?

Thanks for the nice comment; but unfortunately we didn’t try it.

Line 362 “supplemented with 10% (v/v) 1%(w/v)” should be “10% FBS” 

 Thanks for the nice comment; the correction had been made as requested.

Round 2

Reviewer 2 Report

The authors have provided an extensive editing of English language as required. The overall recommendation is: accept in present form.